# Comparative Assessment of Sustainable Consumption Based on the Digital Information Environment Content-Thematic Component Differentiation

**Yanjie Li [1], Daria Terenteva [2], Olga Konnikova [3] and Evgenii Konnikov [4,\***

[1] International Business School, Shaanxi Normal University, Xi'an 710119, China; liyanjie0610@snnu.edu.cn
[2] Faculty of Sociology, St Petersburg University, 199034 Saint Petersburg, Russia; daria_terenteva@inbox.ru
[3] Department of Marketing, Saint-Petersburg State University of Economics, 191023 Saint Petersburg, Russia; olga.a.konnikova@gmail.com
[4] Graduate School of Industrial Economics, Peter the Great St. Petersburg Polytechnic University, 195251 Saint Petersburg, Russia
[*] Correspondence: konnikov.evgeniy@gmail.com; Tel.: +7-9618084582

**Abstract:** The modern culture of consumption often does not correspond to the principles of sustainable development, which leads to environmental pollution. As a possible solution to this problem, the authors propose to analyze the concept of sustainable consumption, following which can reduce the negative impact of mankind on the environment, which is an urgent issue at the global level. The aim of the study is to build a tool for comparative analysis of the level of sustainable consumption. The analysis is carried out in the digital environment, since this space provides a relative freedom of expression of the opinion of individuals, and the obtained data reflect the real level of "sustainability" of the behavior of participants in thematic social communities. The result of the research is an analytical tool tested on thematic social communities of the VK.com social network. In the process of building an analytical tool, primary and secondary tokenization were carried out using computer linguistics tools, an analytical dataframe was formed for a source reflecting the studied content-thematic component. To automate the process of forming a set of tokens, Python 3 programming language was used. The proposed tool can be used to determine the level of sustainable consumption in different social communities relative to each other. This approach can be used by companies to find target audiences and to promote environmentally friendly products. Moreover, this tool can be applied to attract consumers by increasing the level of sustainability in the corporate culture based on the research results. A separate problematic of realizing the developed methodology is the ethical component of analyzing natural digital information and the use of the results obtained on the basis of its analysis.

**Keywords:** sustainable consumption; information environment; content-thematic component; tonal component

## 1. Introduction

In the modern world, consumer culture [1] is progressing every year due to the availability of various material goods. At present, there is a tendency to buy as many goods as possible, regardless of the degree of their necessity for an individual [2]. Thus, people meet their consumer needs by focusing on the process of obtaining different products, rather than on their use. One of the key factors affecting this situation is the development of the advertising sector in terms of psychological impact on the individual [3] and technological progress in visual culture. In addition, the capitalist system in many countries promotes the idea of equivalence between "success" and "profit", and the availability of funds can be reflected through the acquisition of as many goods as possible. Taken together, this leads to the widespread adoption of this pattern of consumer behavior.

As a consequence of this approach to consumption, the world community deviates from the principles of sustainable development [4], in particular, damaging the environment.

At present, environmental issues are among the top priorities at the global level. This is due to environmental degradation and such problems as ocean plastics pollution [5], climate change [6], and many others. Infrastructure and technology are developing, but these changes threaten the environment, and excessive consumption only exacerbates the situation. Thus, it is necessary to reduce the environmental impact by all available means, including the transition to sustainable consumption.

Sustainable consumption as a concept means that an individual acquires and uses only the necessary and sufficient quantities of products, thus, reducing waste and choosing the most environmentally friendly goods and services [7]. As demand for sustainable products increases, manufacturers are also forced to move towards sustainable production, thereby, reducing their impact on the environment. If this concept is used universally at all levels, from individual to state level, it will have a significant positive impact on the environment.

Sustainable consumption is now gaining popularity in countries that give high priority to followings the concept of sustainable development, such as Sweden. If any action is to be taken to increase the level of sustainable consumption, its initial value needs to be determined at the individual level.

This study analyzes the information flow in thematic communities in social networks. Since the concept of sustainable consumption is based on the principle of awareness, public discussion of this topic is also a manifestation of awareness and can be interpreted as the indicator of a sustainable consumption level.

The problem the author intends to address in the research is to determine the level of sustainable consumption. This problem is really important and relevant because the consumption of goods is an integral part of human life that can be changed so that the environmental impact is reduced. Moreover, environmental issues are becoming more and more important every year due to the deteriorating environmental situation.

It is necessary to separately highlight the problem of confidentiality of personal data. The proposed methodology assumes removal from the category of personality and transition to the category of the aggregate of individuals. The data analyzed in the framework of such studies do not allow using the results for the purposes of influencing specific individuals. Moreover, all analyzed information has open access. Despite these facts, the significant development of such research methods presupposes the constant strengthening of the possible problems related to the confidentiality of personal data. In our opinion, the formation of a single international convention regulating the ethical rules for the analysis of natural digital information and the use of its results is extremely significant for the development of such methods.

The aim of the study is to build a tool for comparative assessment of sustainable consumption and propose activities to improve this indicator.

The article consists of introduction, literature review, research methodology and results, discussion and conclusion. In the introduction, the rationale for the relevance of the study is presented, the problem and the aim of the study are indicated. In the literature review, the authors highlight the role of the ecological component in the concepts of sustainable development and sustainable consumption, as well as analyze their relationship and interdependence from each other. In the Research Methodology section, the authors formulate the research question: how can we compare the level of sustainable consumption of different subjects? The main hypothesis is: the results of sustainable consumption are reflected in the objects of the information environment and their transformation. This hypothesis was tested by applying the method of parsing and analyzing natural digital information in thematic communities in social media. The research results section presents the constructing of a tool for comparative assessment of the level of sustainable consumption of participants in three student communities in the Russian social media VK.com, as well as the results of quantification of the level of sustainable consumption of individual subjects by analyzing the information environment of their concentration. In the Conclu-

sion section, main conclusions are formulated based on identifying the links between the study carried out by the authors and the research of scholars analyzed in the framework of the literature review, and also the directions of future research on the topic are presented.

## 2. Literature Review

Sustainable consumption refers to the use of services and related products, which respond to basic needs and bring a better quality of life while minimizing the use of natural resources and toxic materials as well as the emissions of waste and pollutants over the life cycle of the service or product so as not to jeopardize the needs of future generations. The concept of sustainable consumption was formulated at the Oslo Symposium in 1994 [8].

This trend has been widely adopted in today's world as it contributes to the achievement of the sustainable development goals. The Sustainable Development Goals (SDGs) or Global Goals are an ambitious package of 17 global goals intended to be achieved by 2030, establishing a new paradigm of economic and social development, including many goals related to solving environmental problems [9].

The idea of "sustainability" as a development direction was first formulated in the 1987 report Our Common Future (also known as the Brundtland Report) by the World Commission on Environment and Development (WCED), published by the United Nations through the Oxford University Press [10]. The report defined sustainable development as development that meets the needs of the present without compromising the ability of future generations to meet their own needs. The UN Conference on Environment and Development in Rio de Janeiro in 1992 adopted the provisions of sustainable development and formulated actions for its implementation [11].

Sustainable development is a qualitative approach that covers economic and social spheres as well as the environment. Thus, it includes political aspects aimed at improving the social environment and achieving "equal opportunities" for all citizens, as well as environmental and economic measures aimed at efficient use of resources and environmental damage reduction.

Sustainable consumption is one of the aspects of sustainable development, the one public attention is focused on today. This is due to the development of consumerism culture and increased consumption, as these interrelated phenomena lead to increased environmental damage. Unsustainable consumption is mainly characteristic for developed countries, as it is in these countries that a significant proportion of various services and goods acquisition meets people's consumerist consumption needs rather than the actual use of goods [12]. However, there is also a rising trend of "unsustainable" consumption in developing countries, particularly in the domestic sector. Studies show that consumption in the European Union household sector directly or indirectly—through the use of services and goods—accounts for approximately 70% of the total environmental impact.

Sustainable consumption cannot be considered in isolation from concepts of "economic sustainability" and "environmental sustainability". According to the European Commission, the Sustainable Development Strategy calls for the "integration of economic, social and environmental considerations so that they are coherent and mutually reinforce each other" [13].

Sustainable consumption from an environmental point of view is less harmful to the environment, which in turn cannot but affect the value component of various products or services and, accordingly, consumer behavior.

When we focus on the environmental aspects of sustainable development, we look at natural resources, both renewable and non-renewable, that make up our environment and help us maintain and improve our lives.

Environmental issues are inextricably linked to economic issues such as poverty. People living in poverty can wreak havoc on the environment as they struggle simply to survive, chopping trees for firewood, depleting agricultural land, and polluting urban water supplies with waste they cannot afford to process.

Environmental issues are also linked to social issues such as population growth. The rapidly growing population puts pressure on the country's natural resources, as well as its ability to provide housing, health care, education, safe water, and sanitation for everybody [14].

There is a theory of Green Growth or ecological modernization, actively promoted by the Organization for Economic Cooperation and Development (OECD), the World Bank, the UN Environment Program, as well as the EU and its member states since 2011 [15]. Green growth means fostering economic growth and development while ensuring that natural assets are preserved and that they continue to provide ecosystem services on which our well-being depends. Proponents of green growth believe that achieving environmental goals, including mitigating the effects of climate change, will require much more government actions than in previous decades, setting goals and targets, managing risks, promoting industrial policies, realigning prices and countering negative business interests.

Another theory, the purpose of which is to achieve sustainability, is the theory of anti-growth [16]. It is a socio-economic concept that asserts the need to reduce the size of the economy to ensure social welfare in the long term. Unlike a recession in the growth-oriented economy, anti-growth implies a targeted economic and social transformation to maximize the level of happiness and well-being because the time freed up by reducing personal consumption and efficient organization of social labor is devoted to the arts, music, family, culture, and the community. The key point for anti-growth advocates is the recognition of the existence of ecological limits for economic activity. In this regard, the concepts of "ecological capacity" or "ecological footprint" are used.

Supporters of this theory turn to Hartmut Rosa's theory of social acceleration [17]. This theory asserts that the consequences of capitalist thinking and approach to life lead to constant acceleration in various areas. Since the concept of anti-growth allows to radically change the approach to the social and economic model of human behavior in society, this, in turn, will affect the ecological component, as it was described above, these three concepts are interrelated in the context of sustainability.

The main discussion is that if we stick to the capitalist model of behavior, it will only lead to more production and consumption, while sustainable consumption focuses on reducing the share of consumption of goods and services. Schachtschneider supports the theory of anti-growth and proposes to introduce Ecological Basic Income as a solution [18]. The author argues that the social safety net that EBI represents will provide greater freedom of choice among consumers and define the pattern of consumer behavior as more eco-friendly.

However, there is another side to the issue, expressed by van den Bergh [19]. Since all three concepts of "economic sustainability", "environmental sustainability", and "social sustainability" are interconnected, the most controversial link is the economy. The very concept of anti-growth is ambiguous and rather confusing due to significant variations in its interpretation. Moreover, the slowdown in economic growth raises questions.

Despite the fact that the concept of anti-growth is based on the same goal-setting as the concept of sustainable development, its essence is in many ways contrary to the essence of economic development as a whole. A consequence of the implementation of the concept of anti-growth may be a halt in innovative development as a result of decommissioning of key research centers. Thus, technological development that is invariably necessary to increase the sustainability of technological solutions can be replaced by technological degradation, which in turn completely contradicts the concept of sustainable development.

Thus, sustainable consumption cannot be considered in isolation from the concepts of economic and social sustainability in the context of the environmental aspect.

Thus, in today's world with a dominant consumerist culture it is necessary to adhere to the principles of sustainability to prevent significant environmental damage [20]. Each of the considered categories of everyday life affects the global environmental problems of our time, therefore, the introduction of the concept of "sustainability" and consideration of

the level of awareness of the population in this issue are paramount tasks in the context of sustainable development.

## 3. Methodology

The process of sustainable consumption can be characterized by multiple characteristics that can be measured in terms of both quantitative and nominal scales. Quantifiable characteristics reflecting the consumption sustainability degree can include the amount of funds allocated to charity, the share of "eco-friendly" goods in the consumer basket of the research subject and more. However, the most significant characteristics reflecting the consumption sustainability degree are qualitative and can be evaluated only expertly or through other heuristic methods of analysis [21]. Moreover, it should be noted that at the moment, the statistical information collected in the field of sustainable consumption is exclusively of local nature, it is a result of separate sociological and marketing re-search, and in most cases it cannot be universalized and extrapolated for the purpose of forming methodological tools of analysis and management. The analytical approaches and tools considered earlier also cannot be universalized for the purposes of forming methodological level tools. Therefore, the starting point of the process of forming a toolkit to assess the consumption sustainability degree, or level of sustainable consumption, is an exclusively formed general theoretical basis.

As it was established earlier, sustainable consumption should be understood as the process of using goods and services that meet basic needs and improve quality of life while reducing the use of natural resources and toxic materials, as well as waste and pollution throughout the life of the service or product. Thus, we can conclude that the process of consumption is differentiated into two basic components:

1. An increase in the personal benefit of the subject of consumption. This increase is a natural property of any consumption process.
2. An increase in public benefit expressed in reduced use of natural resources and toxic materials, as well as reduced waste and pollution.

These components, provided that they are maximized, are characterized by inverse functional relation. Therefore, sustainable consumption implies the formation of a balance of quantitative interpretations of the complex components' data. As noted earlier, the manifestation of each of the selected components can be expressed through a variety of qualitative and quantitative variables. However, the very properties of sustainable consumption can be manifested within two parallel environments:

1. The objective environment (the real world), which is a sum of material objects and their transformation.
2. The information environment, which is a sum of information objects and their transformation.

Most sustainable consumption studies currently underway are aimed at analyzing the results of the transformation of the real world, such as reducing the amount of environmental and social damage caused by consumption, increasing the availability of natural resources, and so on. The analysis of the resulting parameters, as well as the analysis of the transformation of the characteristics of the real world in general, is associated with many natural (physical) limitations [22]. At the same time, the information environment is much more accessible for analysis. This fact is a consequence of the information society development and the digitalization of major economic and social processes. The mass development of Internet technologies has triggered the digitalization of social communication, resulting in the widespread penetration of online news outlets and social networks. The information environment is not hierarchically managed and centralized, which indicates the potential reliability of information flows generated and transmitted within it. At the same time, the transformations of the real world are inevitably reflected in the information environment. In this study, a hypothesis is put forward that the results of sustainable consumption are reflected in the objects of the information environment and their transformation. There-

fore, quantification of the sustainable consumption level can be implemented through the analysis of the information environment.

The objects of the information environment are elementary information units encoded in textual or audiovisual form and forming the information background, centered on the elements of the real world and comprehensively describing the process of their transformation and interaction. These elementary information units are generated by subjects of the information environment within specialized sources of information concentration (mass media, social networks, thematic and professional communities, etc.). The information environment level of development can be largely characterized by the complexity of this information background. At present, within a certain part of information concentration sources, not only primary information is significant, but also meta-information differentiated into many levels and containing such components as business reputation of the source of generation, business reputation of the overwhelming part of consumers, correlation with the general information context, and more.

It is possible to differentiate both elementary information units and their sets into two basic components:

1. Content-thematic component reflecting multidimensional lexical specificity of an elementary information unit or its aggregate.
2. Tonal component that reflects the properties of the emotional specificity of an elementary information unit or its aggregate.

It should be assumed that identification, structuring, quantification, and comparative analysis of the content-thematic component of an elementary information unit or its set will make it possible to characterize the current properties and the process of its transformation, not only for the complex elements of the information environment, but also for the complex elements of the real world, one of which is the level of sustainable consumption, or the consumption sustainability degree [23,24]. Thus, it is possible to propose a hypothesis according to which, for the purposes of comparative analysis of sustainable consumption level, the basis can have the form of a set of elementary information units selected in accordance with the specifics of the object of study and aggregated into a single array.

Computer linguistic tools, in particular tokenization algorithms, can be used to study a certain content-thematic component of the information environment. As a result of the tokenization process, an array of lexemes is formed, reflecting a certain content-thematic component of the information environment. Thus, the hypothesis put forward earlier can be mathematically interpreted in the form of the coefficient of relative presence of the subject under study (or set of subjects), sustainable consumption in the information background content-thematic component, which can be universally represented in the information environment of the social community. Figure 1 shows a high-level algorithm for estimating the relative presence of a certain content-thematic component in the information background of the subject under study.

The above algorithm assumes parallel formation of a set of tokens for both the information environment content-thematic component and the information environment of the social community, which results in the formation of two arrays of tokens. The quantification of the presence of the content-thematic component in the social community information environment implies the calculation of a previously defined comparative coefficient, which in an isolated form has no analytical value. It obtains the analytical value in case of comparing several objects (social communities).

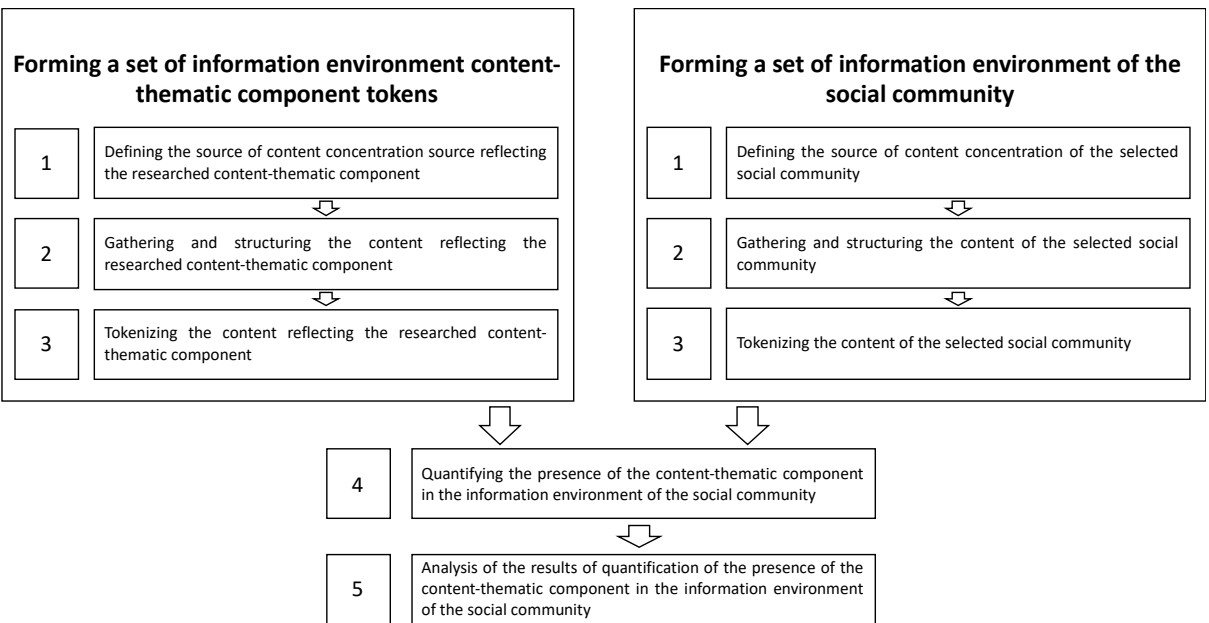

**Figure 1.** A high-level algorithm for estimating the relative presence of a certain content-thematic component in the information background of the subject under study.

For the purpose of instrumental clarification of this algorithm, this study further assesses the level of sustainable consumption by several communities. For the purposes of comparative analysis, three student communities on the social network VK dedicated to sustainable consumption were identified. The first analyzed community is "Green HSE in St. Petersburg" (https://vk.com/spbgreenhse accessed on 20 June 2021), a community of students at the Higher School of Economics, St. Petersburg. There are 1153 subscribers in the community, 79% of which are women, the average age is 23.8 (SD = 0.35) and 76.6% are from St. Petersburg. At this university, mainly students of economic secession study, but there are also programs for designers, lawyers, historians, political scientists, and many others. The second analyzed community is "ITMO.GREEN (https://vk.com/itmo_green accessed on 20 June 2021), a community of students from the ITMO University. There are 1101 subscribers in the community, of which 75.5% are women, with an average age of 24.4 (SD = 0.4), and 82.1% are from St. Petersburg. There are different specializations at the university, but the majority of students study information technology, photonics, robotics, quantum communications, chemistry, and biotechnology. The third analyzed community is "ReGreen" (https://vk.com/regreen_polytech accessed on 20 June 2021), a community of students of Peter the Great St. Petersburg Polytechnic University. There are 1534 subscribers in the community, of which 71% are women, the average age is 24.9 (SD = 0.4), and 81.4% are from St. Petersburg. This is a classic university in which students of a huge number of directions study, although the priority is given to technical sciences. These three communities were selected for the study since these are the largest student communities in Russia on the social network VK.com, dedicated to sustainable consumption and ecology. All three communities are similar to each other in terms of audience size and its socio-demographic profile, significant differences exist only in the predominant specialization of teaching for the students.

This sample results from the fact that these communities are thematically consistent and, therefore, comparable as they focus on one of the basic aspects of sustainable consumption, and because the student part of society is the most active in the information environment. For the purpose of forming a set of tokens of the content-thematic component of the information environment, the "Ecology" community in the VK.com social network (https://vk.com/ecoonf accessed on 20 June 2021) was also selected. This choice is conditioned by the fact that this community is the most active and popular one in the Russian segment of the Internet out of those devoted to the problems of ecology and environmental

protection (its total audience is about 19,000 people, and it hosts at least three thematic posts every day). The specifics and results of the research are given in the next section of this article.

## 4. Results

Python 3 programming language was used to automate the process of forming a set of tokens of the content-thematic component of information environment and social community information environment. The algorithm that automates the primary stages of analysis is shown in Figure 2.

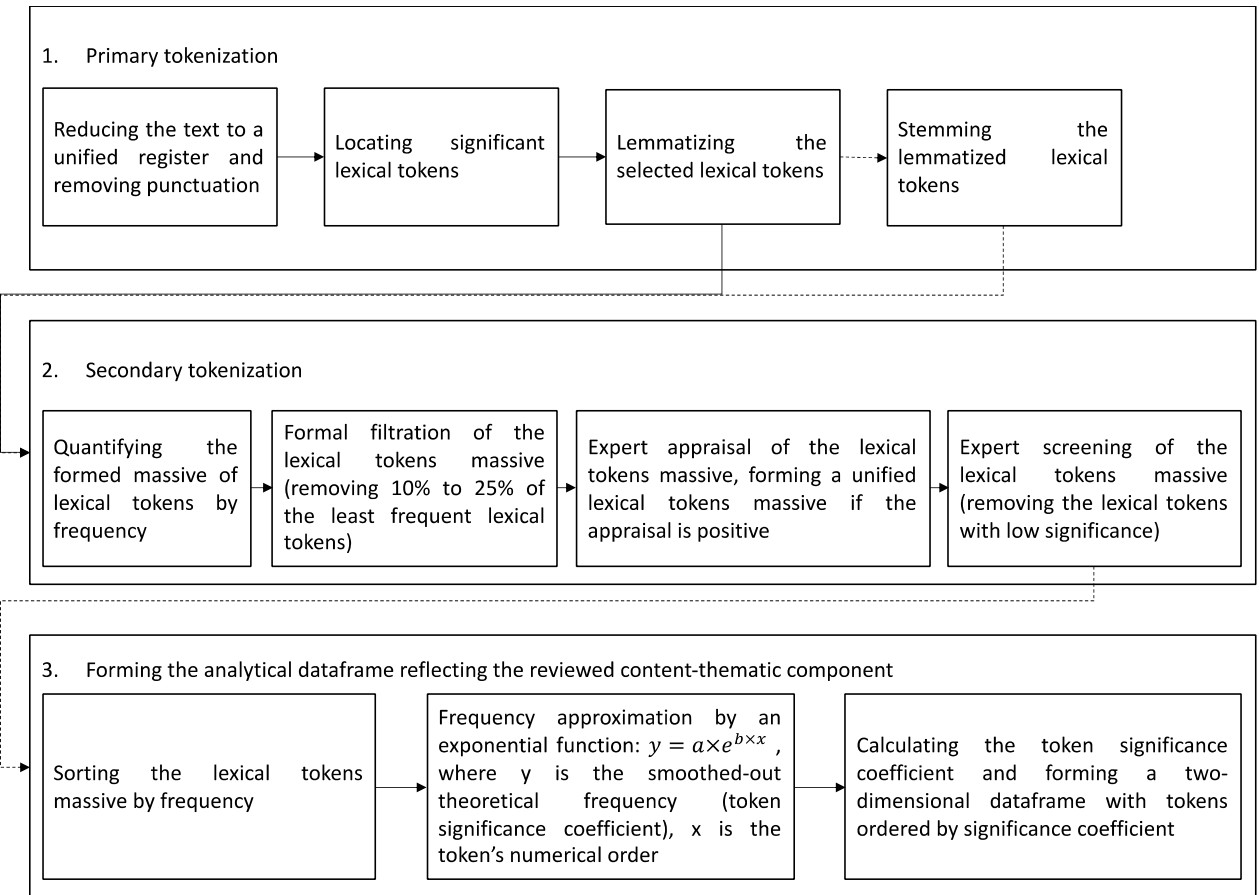

**Figure 2.** Algorithm of forming a set of tokens of the content-thematic component of the information environment and social community information environment.

As can be concluded in accordance with the above algorithm, the primary and secondary tokenization is uniform for both investigated elements, while the third stage of the algorithm—the formation of the analytical dataframe—is implemented exclusively for the source reflecting the researched content-thematic component.

For the purposes of quantification of the presence of a content-thematic component in the information environment of the social community, a universal coefficient—the Content-thematic Component Coefficient (CCC)—has been developed. This coefficient reflects the comparative level of presence of the content-thematic component of the information environment in the unit (or set of units) of the social community information environment, and is defined as follows:

$$CCC = \sum_{1}^{n} T_n \times y_n$$

where: $T_n$ is the sum of references to the n token in a unit (or set of units) of the social community information environment; $y_n$ is a smoothed-out theoretical frequency (coefficient of significance) of the n token.

The study identified 279 tokens for which the smoothing function took the following form:

$$y_n = 39.386 \times e^{0.0119 \times x_n}$$

where: $x_n$ is the numerical order of the n token (tokens are ordered by increasing frequency of reference in the Ecology community information environment).

The determination coefficient of this model is 0.941, which indicates the high quality of the model, the Fisher F-criterion and the *p*-value at $x_n$ indicate the model reliability and high significance of the exogenous variable. Examples of dedicated tokens are shown in Figure 3.

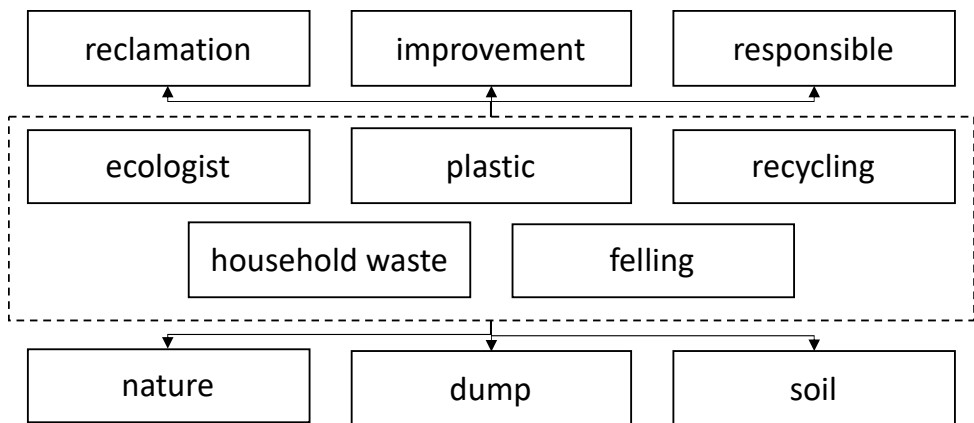

**Figure 3.** Examples of tokens.

As a result of quantification, an analytical dataframe was formed from the evaluation of each individual post. For the purposes of comparative analysis within the framework of this study, it is most effective to use single-factor analysis of variance. Table 1 shows the results of single-factor analysis of variance.

**Table 1.** Analysis of variance.

| Source of Variation | SS | Df | MS | F | *p*-Value | F Critical |
|---|---|---|---|---|---|---|
| Between communities | 32,968,577.78 | 2 | 16,484,288.89 | 5.76 | 0.0032 | 3.004 |
| Within communities | 3,118,333,032 | 1090 | 2,860,855.992 | | | |

It shows that the communities analyzed are significantly different according to the coefficient of the substantial thematic component "sustainable consumption", which is primarily indicated by the *p*-value of 0.0032. Therefore, despite the thematic uniformity, the level of sustainable consumption manifests itself differently in each of the analyzed student communities. A comparison of mean and variance for each of the sample elements is shown in Figure 4.

According to the results presented on the diagram, we can conclude that the com-plex properties of sustainable consumption are most evident in the community of students of the Peter the Great St. Petersburg Polytechnic University—"ReGreen". However, this community is also characterized by the largest variance, which indicates the heterogeneity of its content in terms of the content-thematic component of sustainable consumption presence. The lowest value of the research coefficient is observed for students at the Higher School of Economics, St. Petersburg—"Green HSE in St. Petersburg". The variance is also

relatively high. The lowest level of variance is observed in the community "ITMO.GREEN", which indicates the most stable level of content in terms its sustainable consumption content-thematic component.

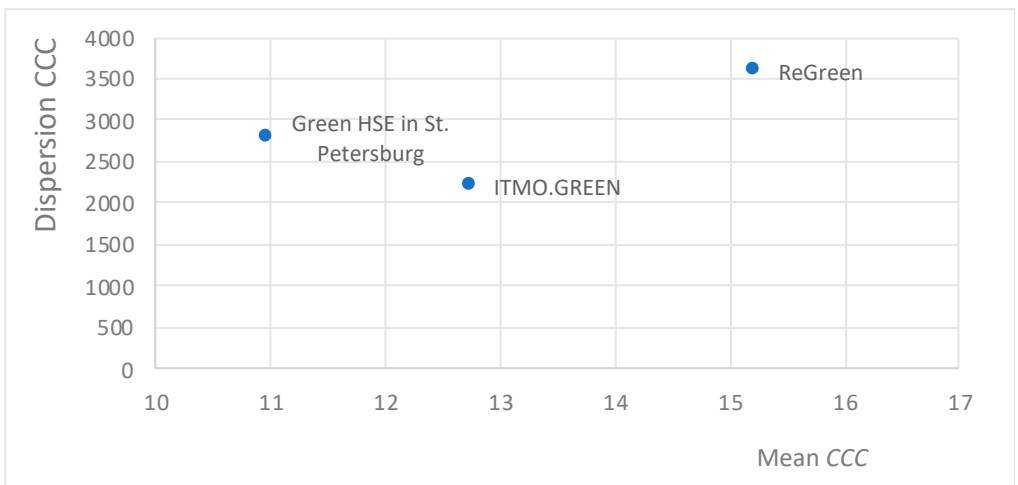

**Figure 4.** Distribution of average and variance of sample elements.

Thus, the tool allows to estimate the level of sustainable consumption, or the consumption sustainability degree, for the purposes of comparative analysis of information subjects or their sets. Based on this comparative analysis, a target segment of managerial or marketing impact can be identified, as well as a set of potential consumers of products that rely on being a part of the sustainable consumption concept as their critical success factor. Therefore, for the purpose of further development of this tool, it is necessary to apply it consistently to different samples and formulate high-level conclusions.

It should be noted, separately, that the resulting indicator primarily characterizes the representation of sustainable consumption topics in the information environment. The presence of the topic of sustainable consumption in the information environment determines the integration of relevant ideas and values into the consciousness of people aggregated within this information environment. This integration of the relevant ideas and values forms the conversion of properties of the information environment into real actions of people, aggregated within the framework of this information environment.

## 5. Discussion and Conclusions

This tool is a promising way to analyze data, as it allows to quantify the information environment in terms of the sustainable consumption concept development. At present, a significant part of everyday life is occupied by the digital environment, and with the beginning of the CoVID-19 crisis its importance has only increased. The opportunity to conduct a comparative analysis in the widespread information space will allow assessing the levels of sustainable consumption relative to each other and eventually to form measures to improve this indicator.

Further use of the tool proposed in this study may apply not only to thematic environmental communities, but also to the official resources of various companies. At present, there is a widespread increase in the level of sustainability in various sectors of the economy, which is associated with global trends in the development of our society. Comparative analysis of social communities that are characterized by belonging to a certain company helps identify trends in the introduction of sustainable development principles into the corporate culture of these companies. Based on the research results, corporations will be able to develop measures to increase sustainability and attract consumers in this way.

Moreover, a comparative analysis of various social communities of potential consumers may show the most promising audience for achieving publicity about environmentally friendly products from the ecologically friendly product producers' point of view.

The proposed tool will help improve the effectiveness of advertising campaigns and, thus, influence the demand for this category of products in a positive way.

This tool can be used most effectively when combined with a preliminary sociological analysis of various social groups of consumers in order to identify means of influence on them in the digital environment in the context of the features of these groups.

To increase the level of sustainable consumption, such measures can be implemented as: environmental actions and activities with full coverage in the official sources of the organizing companies and publicity in thematic social communities. In addition, companies may introduce an environmentally friendly proposition for consumers that would reduce the cost of goods and provide official confirmation of compliance with the principles of sustainable development in the information sources.

Since earlier the hypothesis about the relationship between awareness and sustainable consumption was put forward, it is necessary to implement actions aimed at increasing awareness in a certain audience of the digital space. For this purpose, standard algorithms for promoting and increasing interest in social networks can be used.

In addition, the results of this study can be useful for measuring several quantitative indicators of sustainability, part of which is sustainable consumption in the context of the digital environment. For example, Happy Planet Index (New Economics Foundation (NEF), 2016) was introduced in 2006 (latest data from 2016) to describe human well-being and environmental impacts. The social dimension includes subjective well-being, life expectancy, inequality of outcome and ecological footprint (the average amount of land needed, per head of population, to sustain a typical country's consumption patterns) [25]. Since the resulting indicator characterizes the pattern of consumer behavior, it can be used to assess the complex level of sustainability from the environmental point of view. This is a necessary measure in the context of a critical environmental situation.

In this work, a study of the concept of sustainable consumption was conducted, an algorithm for comparative analysis of sustainable consumption in the context of the information environment was proposed, and recommendations on how to use this tool and increase sustainable consumption were formulated. The subject matter of the study is very relevant at present, due to the difficult environmental situation and trends in implementing the sustainable development concept at various levels of the economic sector.

The aim of the study was to build a tool for comparative assessment of sustainable consumption and propose activities to improve this indicator. This goal was fully achieved by the authors in their work.

In the process of building an analytical tool primary and secondary tokenization were carried out with the help of computer linguistics tools, and an analytical dataframe for the source was formed, reflecting the studied content and thematic component. To automate the process of forming a set of tokens Python 3 programming language was used.

The proposed tool can be used to determine the level of sustainable consumption in different social communities relative to each other. This approach can be used by companies both to promote environmentally friendly products and to attract consumers by increasing the level of sustainability in the corporate culture based on the research results.

The obtained results are extremely important in practical implementation, primarily for companies promoting and capitalizing the values of sustainable consumption, as well as for the government structures in the framework of increasing the level of sustainable consumption of citizens. The applied interpretation of the results can be centered around the thesis that consumers, whose mean CCC level is comparatively higher, can be influenced by the beneficiaries through traditional marketing tools. This thesis is based on understanding the relative propensity of these consumers to sustainable consumption, in comparison with alternative consumer groups identified. For consumers whose mean CCC level is comparatively lower, the beneficiaries should first of all impact the increase of this indicator, gradually promoting the values of sustainable consumption within the group of consumers and, as a consequence, increasing their mean CCC level. As a result, consumers whose mean CCC level was comparatively lower will move to the first consumer group

(with higher mean CCC level) and the beneficiaries will be able to influence them through traditional marketing tools. Moreover, the beneficiaries may take into account the CCC dispersion ratio. This indicator determines the direction of the impact. The bigger the dispersion of the CCC is, the greater the impact should go through related topics in which the group members are most interested. For consumer groups with a comparatively low CCC dispersion ratio the beneficiaries have the ability to influence directly, speaking about sustainable and ecological issues.

One of the key limitations for the implementation of the presented methodology is the exceptional comparative specificity of the results. Since the evaluated criterion is based on the processing of natural digital information, the identifiable tokens characterizing the corresponding content-thematic components are fully limited by the specifics of the generation subjects. Consequently, the obtained absolute values of the presence of the corresponding content-thematic component in the information environment cannot be interpreted directly. Moreover, it should be noted that the possible pre-filtering of the analyzed content, as well as the differentiation of cultural contexts, may cause distortion of the results.

In conclusion, the ethical ambiguity of the application of such a methodology should be noted. The analyzed data is generated by people without realizing the possibility of subsequent analysis. The dissemination of such methods of analysis should inevitably be accompanied by the development of international convention governing the ethical rules for the analysis of natural digital information.

The level of sustainable consumption can be increased by focusing on consumer awareness, advantageous eco-products offers, and digital coverage of environmental events and activities.

**Author Contributions:** Conceptualization, O.K. and Y.L.; methodology, E.K.; software, E.K.; validation, D.T.; formal analysis, D.T.; investigation, Y.L.; resources, E.K.; data curation, D.T.; writing—original draft preparation, O.K.; visualization, E.K.; supervision, O.K.; project administration, Y.L. All authors have read and agreed to the published version of the manuscript.

**Funding:** This research received no external funding.

**Institutional Review Board Statement:** Not applicable.

**Informed Consent Statement:** Not applicable.

**Acknowledgments:** The research was supported by the Peter the Great St. Petersburg Polytechnic University.

**Conflicts of Interest:** The authors declare no conflict of interest.

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
