# Peer review of "Comparative Assessment of Sustainable Consumption Based on the Digital Information Environment Content-Thematic Component Differentiation"

_sustainability, doi:10.3390/su13137215_

Round 1
Reviewer 1 Report
This paper is strongly ideologically biased, taking a position of "moral high ground" with relation to an unsustainable "capitalistic system" and "culture of unsustainable consumerism".
The aim of the paper focuses on the analysis of the linguistics patterns in the open communications of several social communities, in order to make an assessment of the sustainability habits of their culture.
For one part the analysis of what is sustainable and what is not, is incomplete and culturally biased. Stays on the surface without going deep on every option and its sustainability implications for each situation, or aspects of price and affordability for the community. Just plays the "blame game" on the unsustainable consumer. Further on does not concern, for example, with aspects of service based economy vs product based economy, or possible efficiencies generated by technology with its foreseable evolution.
Privacy aspects of human subjects of this study are completely disregarded, and the ethical implications of building an algorithm to build a sustainability "language" judge, and the consequences for the culture and freedom of speech can be dire.
Author Response
Dear reviewer, we are extremely grateful for your comments. In the first version of this article, we really did not accurately describe what sustainable consumption is manifested in, as well as the specifics of economic ties. Also, we totally agree with you regarding the issues of confidentiality and research ethics, and therefore we consider it necessary to form a unified ethical conversion of such research. Based on your comments, we have greatly revised the article, in particular:
- We separately considered the issue associated with the problem of the ethics of such types of research.
- We specified the literature review and generally described the economic specifics.
- We completed the research methodology.
Once again, we thank you for your comments. The methodology developed by us is primarily consumer-oriented, which is currently more clearly expressed in the text of the article. In the revised version of the article, new parts are highlighted in red.

Reviewer 2 Report
The paper aim is very interesting and original.
However, some changes and integrations are needed to make the paper publishable.
I suggest improving the introduction. In particular it is necessary to add, at the end of the introduction, the paper structure, with a small summary of each paragraphs.
The literature review is complete. A little doubt about the source/s of the following statement:
"Considering sustainable consumption in the context of daily life, the following areas of highest impact on the environment can be highlighted:
1. Consumption of food and beverages
2. Residential consumption
3. Transport mobility"
The methodology used is very original, however the composition of the sample and how to identify student behavior are unclear.
I recommend improving the discussion. It is important to recall some of the sources used in the literature. The theoretical implications should emerge from the discussion.
Practical implications and limitations of the work should emerge more clearly in the conclusions.
I suggest making one paragraph "Discussion and conclusions".
I suggest, finally, to add an Appendix contained the quantitative aspects of the empirical research.
Author Response
Dear reviewer, thank you very much for your comments. After being revised, the article became much more meaningful. According to your review, we have completed the following revisions:
- We have added article structure and summaries of each paragraph at the end of the introduction section.
- We have adjusted the literature review and made it more structured.
- We have supplemented the methodology with sample properties.
- We have significantly expanded the discussion section and supplemented it with theoretical conclusions, and also combined it with the conclusion section.
Unfortunately, we cannot add a dataframe, since it is extremely voluminous (more than 2000 elements) and is more than 10 pages. However, if you wish, we could send it to you by e-mail.
Thank you very much, after revising the article based on your comments, this article has become much better.

Reviewer 3 Report
In this manuscript, the authors propose the use of an automated algorithm to characterize and compare the sustainability of consumption habits in various students' communities, via the analysis of their interactions in a social network. Studying the information objects as a way to infer trends and behaviours in the real world is a potentially effective and efficient approach that could be generalized, as the authors suggest, in market research and sociological resarch.
Interesting as it is, the manuscript is also flawed in many senses:
- The introduction and the literature review contain many gross generalizations. I understand this is not a technical paper, but, anyway, unsustainable use of natural resources, climate change, pollution and plastics in the ocean cannot be simplistically equated. This trend to oversimplification permeates all the theoretical first sections. Unless this section is nuanced, it would be better to talk only about depletion of the natural resources, which is worrysome enough to justify the need of the research.
- LIkewise, overly simplistic views are taken in the description of the potentially unsustainable areas of household consumption. E.g., in waste production only recycling is selected, when it's the least impactful of the 3Rs; food consumption is discussed without reference to the change in the share of meat in the diet, only to give a few examples. I would suggest summarizing most obvious pieces of information and deepening in a bit more ambitious alternatives.
- There are details lacking in the description of the sample that allow for understanding better the context, or to make sense of the differences found. The instrument is described in great detail, but its real meaning is quite obscure: what qualitative differences exist among the three samples that justify the difference in the scores, which kind of tokens were found and considered in the analysis, what do these scores mean in practical terms (of behaviours or attitudes)?
- This information lacking, the discussion and conclusions are superficial and of little help. The algorithm seems to be successful because it has been able to identify three distinct communities, but we don't know what are the real differences underlying this scoring. As a result, the suggestions for improvement are too general and of little help.
All in all, I would recommend the authors reformulate the introduction removing some common places and increasing precission, to reduce the technical details (maybe moving them to the supplementary materials section), and to illustrate the results with qualitative information that help the reader understand the meaning of the scores and their implications.
Author Response
Dear reviewer, thank you very much for your comments. After revising the article based on your comments, it has become much more meaningful. According to your review, we have completed the following revisions:
- We rewrote the literature review and introduction sections, specifying the specifics of the perception of sustainable consumption in the framework of our study.
- We dialyzed the description of the sample and the differences between its parts.
- We have detailed the specifics of tokens and provided examples of allocated tokens.
- We have significantly expanded and concretized the discussion section and supplemented it with theoretical conclusions.
In the revised version of the article, the new parts are highlighted in red.

Round 2
Reviewer 1 Report
Dear authors, the paper is much improved with the modifications you added.
A section about the criticisms of the anti-growth theory should be added. Plus the paper makes a connection with anti-growth and UBI wich is not clear. While UBI is a measure to face unequality it is not clear its connnection with anti-growth.
Author Response
Dear Reviewer, thank you so much for your comment. Indeed, in the process of finalizing the article, we missed a moment with criticism of the concept of anti-growth. We have added a meaningful section criticizing this concept.

Reviewer 2 Report
publishable in present form
Author Response
Dear Reviewer, thank you so much for your approval of our paper.
Reviewer 3 Report
Although the introduction section has been improved and contains now less imprecise pieces of information, the addition of some other sections is not so adequate. For example, I cannot see how the discussion of mathematical roots of the concept of "sustainability", which identifies it with stability, contributes to the discourse around environmentally and socially bearable development.
Although the methods are now clearer (some examples of tokens have been included, which makes it easier to understand the approach, the communities are described), it is still hard to see how the different scores across communities result in qualitative differences or, in other words, what pratical implications do they have.
Author Response
Dear Reviewer, thank you so much for your comment. Indeed, in the process of finalizing the article, we've forgotten to describe the specifics of practical implementation of the developed tools. We have added a separate paragraph about this.

Round 3
Reviewer 3 Report
I think the paoer is now ready for publication
Author Response
Thank you very much!